# Localized Data Shapley: Accelerating Valuation for Nearest Neighbor Algorithms

**Guangyi Zhang**
Shenzhen Technology University
zhangguangyi@sztu.edu.cn

**Yanhao Wang***
East China Normal University
yhwang@dase.ecnu.edu.cn

**Chengliang Chai**
Beijing Institute of Technology
ccl@bit.edu.cn

**Qiyu Liu**
Southwest University
qyliu.cs@gmail.com

**Wei Wang***
HKUST(GZ) and HKUST
weiwcs@ust.hk

## Abstract

Data Shapley values provide a principled approach for quantifying the contribution of individual training examples to machine learning models. However, computing these values often requires computational complexity that is exponential in the data size, and this has led researchers to pursue efficient algorithms tailored to specific machine learning models. Building on the prior success of the Shapley valuation for $K$-nearest neighbor (KNN) models, in this paper, we introduce a localized data Shapley framework that significantly accelerates the valuation of data points. Our approach leverages the distance-based local structure in the data space to decompose the global valuation problem into smaller, localized computations. Our primary contribution is an efficient valuation algorithm for a threshold-based KNN variant and shows that it provides provable speedups over the baseline under mild assumptions. Extensive experiments on real-life datasets demonstrate that our methods achieve a substantial speedup compared to previous approaches.

## 1 Introduction

Data has emerged as the new oil of the digital economy, driving advances across various fields such as artificial intelligence, healthcare, finance, and beyond. The rapid growth in data collection has created unprecedented opportunities for building powerful machine learning models that can solve complex problems. As organizations increasingly rely on data-driven decision making, the quality and relevance of training data have become a critical factor that determines the success of machine learning applications and consequently impacts different aspects of our daily lives [1]. This paradigm shift has highlighted the fundamental importance of data as a valuable resource that requires careful management, curation, and valuation [2].

Despite the widespread recognition of the importance of data, a significant challenge remains: how to systematically, fairly, and efficiently value individual data points within large datasets. This question is increasingly relevant due to the emergence of data marketplaces, the evolution of privacy regulations, and the development of data-centric AI. Data valuation serves multiple purposes, including compensating data contributors equitably, identifying high-value data for acquisition, and removing harmful or misleading examples [3–5].

The Shapley value (SV), originating from cooperative game theory, is a principled and theoretically grounded approach to addressing the challenge of data valuation [6]. Shapley values provide a

---

*Corresponding authors.

39th Conference on Neural Information Processing Systems (NeurIPS 2025).

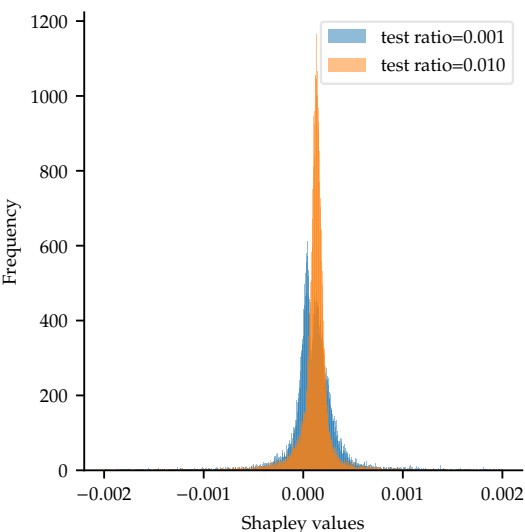

Figure 1: Illustration of the distribution of Shapley values computed for different test data ratios.

mechanism for fairly distributing the collective value of a coalition among its members. When applied to machine learning, data Shapley values [7, 8] measure the contribution of each training example to model performance by considering its marginal value across all possible subsets of the dataset. This approach satisfies important properties such as fairness, additivity, and symmetry, making it a reliable measure for data valuation tasks. Note that it differs from the attribution methods such as SHAP [9], where Shapley values are computed for each feature of a given data point.

However, the computation of data Shapley value is expensive, as it requires enumerating all possible subsets of data points. In fact, it has been shown to be #$\mathbf{P}$-hard in certain games [10]. A recent breakthrough by Jia et al. [11] exploits the structure of unweighted $K$-nearest neighbor models (also known as KNN models) to efficiently compute the *exact* data Shapley values. KNN models are a family of classic machine learning models that predict the label of a data point based on the labels of its $K$ nearest neighbors. KNN models can be adapted to modern neural networks by using their learned embeddings as the feature space. The KNN-based Shapley value (KNN-SV) has quickly become one of the leading data valuation techniques [12–15].

Although KNN-SV offers significant speed improvements over the naive approach, it still struggles with large datasets due to its linear dependence on the test data size. In KNN-SV, a value attributed to each test data point is distributed among all training examples based on their contribution to the prediction. Owing to the local nature of KNN models, an average contribution over a sufficient number of test points is required for an accurate valuation. See Fig. 1 for an illustration of the data values computed for different numbers of test points on a real dataset. When the number of test points is small, the Shapley values have a more dispersed distribution, indicating that the values are less stable and more extreme. Hence, it is desirable to use a number of test points that is of the same order of magnitude as the training set size, but this will result in quadratic overall time complexity.

In this paper, we address this key limitation of prior KNN-SV approaches and propose a more efficient algorithm. Our algorithm leverages the distance-based local structure in the data space to decompose the global valuation problem into smaller, localized computations. For a threshold-based KNN variant, our algorithm provides provable speedups over the baseline under mild assumptions. We validate the proposed algorithm through experiments on real-world datasets, demonstrating practical improvements over existing methods.

The remainder of the paper is organized as follows. We first review the preliminaries in Section 2 and present a baseline method that computes KNN-SV by recursion in Section 3. We then present our proposed methods by starting with landmark-based near neighbor search in Section 4 as a building block and introducing a fast method for threshold-based KNN in Section 5. The experimental setup and results are outlined in Section 6. We conclude the whole paper in Section 7.

## 2  Preliminaries

In this section, we introduce the framework for data valuation based on the Shapley value (SV) and establish our notation for applying this concept to $K$-nearest neighbor (KNN) models.

### 2.1  Cooperative Game Theory and Shapley Value

The concept of data valuation can be elegantly formalized through the lens of cooperative game theory. In this framework, we consider a collection of players who can form coalitions to generate collective utility. Formally, a cooperative game consists of a pair $(I, v)$, where $I = \{1, \ldots, n\}$ represents the set of players and $v : 2^I \to \mathbb{R}$ is a utility function that assigns a real value to each possible coalition.

A central question in cooperative game theory concerns fair allocation: How should the total utility be distributed among individual players based on their contributions? The Shapley value, introduced by Lloyd Shapley [6], provides a time-tested solution to this problem. For each player $i$, the Shapley value $s(i)$ represents the average marginal contribution across all possible coalition formations, i.e.,

$$s(i) = \frac{1}{n} \sum_{S \subseteq I \setminus \{i\}} \binom{n-1}{|S|}^{-1} [v(S \cup \{i\}) - v(S)].  \tag{1}$$

The Shapley value *uniquely* satisfies the following four desirable properties, which makes it particularly suitable for data valuation. That is, there does not exist any other value function that can simultaneously satisfy all of them. (1) **Efficiency**: The total utility is completely distributed among all players, i.e., $v(I) = \sum_{i \in I} s(i)$. (2) **Symmetry**: Players with identical marginal contributions receive equal value, i.e., if $v(S \cup \{i\}) = v(S \cup \{j\})$ for all $S \subseteq I \setminus \{i, j\}$, then $s(i) = s(j)$. (3) **Null Player**: Players who contribute nothing to any coalition receive zero value, i.e., $s(i) = 0$ if $v(S \cup \{i\}) = v(S)$ for all $S \subseteq I \setminus \{i\}$. (4) **Linearity**: Values under multiple utility functions sum to the value under the combined utility, i.e., $s_{v_1}(i) + s_{v_2}(i) = s_{v_1+v_2}(i)$ for all $i \in I$.

In the context of machine learning, we can reinterpret players as individual data points in the training set and the utility function as a performance measure of models trained on different subsets of the data. This naturally leads to a framework for quantifying the contribution of each training data point to the overall model performance.

### 2.2  KNN-Based Shapley Value (KNN-SV)

In this subsection, we introduce the KNN-based Shapley values with respect to two different utility functions, one for the standard KNN classifier and the other for a threshold-based KNN classifier. These utility functions are specifically targeted for a single test data point, and we conclude with a discussion on how to extend them to multiple test points, which are required for an accurate and balanced valuation.

Given a dataset $D$ of size $n$, where $z = (x, y) \in D$ with $x \in \mathbb{R}^d$, $y \in \mathcal{Y}$, and $\mathcal{Y}$ is the label space, we want to compute the Shapley value $s(z \mid z_{\text{test}})$ of $z$ with respect to a test point $(x_{\text{test}}, y_{\text{test}})$. Let the weight of $z$ be $w(z \mid z_{\text{test}}) \in \mathbb{R}_+$ that indicates the proximity between $x$ and $x_{\text{test}}$ or is simply a constant in the case of an unweighted KNN. Following the formulation in [11], let $v$ be the weighted KNN utility function. We have

$$v(S) = \sum_{i=1}^{\min(|S|, K)} w(z_{\alpha_i(S)} \mid z_{\text{test}}) \, \mathbb{1}(y_{\alpha_i(S)} = y_{\text{test}}),  \tag{2}$$

where $\alpha_i(S)$ is the index of the $i$-th closest element of $S$ to $x_{\text{test}}$, and we call $i$ the *rank* of $z_{\alpha_i(S)}$ in $S$. When $w(z \mid z_{\text{test}})$ is a constant, e.g., $w(z \mid z_{\text{test}}) = 1/K$, the utility function $v$ is derived from the standard unweighted KNN classifier. On the other hand, if we write the distance as $\mathsf{d}(z, z') = \|x - x'\|$, and let the weight of $z$ be the Gaussian kernel as

$$w(z \mid z_{\text{test}}) = \mathcal{K}(\mathsf{d}(z, z_{\text{test}})) = \exp(-\mathsf{d}(z, z_{\text{test}})^2 / 2\sigma^2),$$

where $\sigma$ measures the *width* of the Gaussian kernel, the utility function $v$ is derived from the weighted KNN classifier instead. Note that one is free to use other metric distances other than the Euclidean distance.

We further consider a similar utility function $\bar{v}$ for a variant of the standard KNN classifier, where the utility of a subset $S$ takes into account only the nearest $K$ points to $z_{\text{test}}$ that are within a ball of radius $\tau$ centered at $z_{\text{test}}$. Formally, the utility function is defined as

$$\bar{v}(S) = \sum_{i=1}^{\min(|S_\tau|, K)} w(z_{\alpha_i(S_\tau)} \mid z_{\text{test}}) \mathbb{1}(y_{\alpha_i(S_\tau)} = y_{\text{test}}), \tag{3}$$

where $\alpha_i(S)$ is the index of the $i$-th closest element of $S$ to $z_{\text{test}}$, $S_\tau(z_{\text{test}}) = \{z \in S \mid \mathsf{d}(z, z_{\text{test}}) \le \tau\}$, and we write $S_\tau = S_\tau(z_{\text{test}})$ for short. This variant is derived from a more robust threshold-based KNN classifier (TKNN) [14, 16, 17], where a point that is too far away from $z_{\text{test}}$ always has zero influence on $z_{\text{test}}$. It has been shown that this variant possesses additional desirable privacy-friendly properties [14].

With the utility functions defined above, the definition of the Shapley value of a data point $z \in D$ is straightforward, following Eq. (1). Formally, the Shapley value of a data point $z \in D$ with respect to a given test point $z_{\text{test}}$ is defined as

$$s(z \mid z_{\text{test}}) = \frac{1}{n} \sum_{S \subseteq D-z} \binom{n-1}{|S|}^{-1} [v(S+z) - v(S)], \tag{4}$$

where we write $D - z = D \setminus \{z\}$ and $S + z = S \cup \{z\}$ for convenience.

In practice, accurately measuring data values requires multiple test points, which typically increases proportionally with the size of the dataset, $n$. Suppose that there are $n_{\text{test}}$ test points in $D_{\text{test}}$, and the data Shapley value of a data point $z$ can be naturally extended as the average over all test points, i.e.,

$$s(z) = \sum_{z_{\text{test}} \in D_{\text{test}}} s(z \mid z_{\text{test}}) / n_{\text{test}}. \tag{5}$$

## 3 Baseline: KNN-SV by Recursion

In this section, we introduce the analytical solution to the data Shapley values for KNN models, introduced in [11], and extend it to the utility functions in Eqs. (2) and (3). This results in a dramatic improvement in time complexity, from $O(2^n)$ to $O(n \log n)$ for a single test point, over the naive approach that enumerates all possible subsets $S$ of the dataset $D$.

We consider a fixed test point $z_{\text{test}}$ throughout this section. Given a subset $S \subseteq D$, recall that $\alpha_i(S)$ is the index of the $i$-th closest element of $S$ to $z_{\text{test}}$. When the context is clear, for simplicity, we write $z_{\alpha_i(D)}$ as $z_i$ and $w(z_{\alpha_i(D)} \mid z_{\text{test}})$ as $w_i$. Similarly, we denote by $s_i$ the Shapley value $s(z_i \mid z_{\text{test}})$ for the data point $z_i$.

We first restate a known result about the pairwise difference of the KNN Shapley values in the following lemma.

**Lemma 1** (Jia et al. [11]). *Fixing a test point, for any $i, j$, we have*

$$s_i - s_j = \frac{1}{n-1} \sum_{S \subseteq D - z_i - z_j} \frac{v(S + z_i) - v(S + z_j)}{\binom{n-2}{|S|}}.$$

Based on Lemma 1, we can develop a recursive formula for $s_i$ in the following theorem.

**Theorem 2.** *Fixing a test point, for any $i < n$, we have*

$$s_i = s_{i+1} + \min(K, i) \frac{w_i \mathbb{1}(y_i = y_{test}) - w_{i+1} \mathbb{1}(y_{i+1} = y_{test})}{i} \quad and \quad s_n = \frac{K}{n} w_n \mathbb{1}(y_n = y_{test}).$$

See proof in Appendix A. In summary, in order to compute the Shapley values of all data points in the dataset $D$, we start with $s_n$ for the farthest data point and then iteratively apply the recursive formula in Theorem 2 to compute the values of $s_{n-1}, \ldots, s_1$, in decreasing order of their distance to the test point, one data point at a time.

It is easy to see that the recursive formula formed by Theorem 2 also works for threshold-based KNN models. The only difference is that the recursion is applied to the set of data points that are within the radius $\tau$ of the test point, instead of the entire dataset $D$.

**Time Complexity.** We have described a recursive formula for the data Shapley values when there is only a single test point. The calculation requires no more than one sorting of the data points by their distance to the test point, which takes $O(dn + n \log n)$ time. This is a drastic improvement over the naive approach that enumerates all possible subsets $S$ of $D$, whose time complexity is $O(2^n)$.

However, when considering multiple test points, whose size $n_{\text{test}}$ is often in proportion to the size of the dataset $n$, i.e., $n_{\text{test}} = \Omega(n)$, one sorting for each test point amounts to a time complexity of $O(dn_{\text{test}}n + n_{\text{test}}n \log n)$, which becomes quadratic in the size of the dataset. This is clearly too slow for large-scale applications, especially when $n_{\text{test}}$ is large.

## 4   Landmark-based Near Neighbor Search

In this section, we introduce a core building block for our proposed methods, namely landmark-based near neighbor search, which allows to effectively and efficiently shrink the search space of the near neighbors in data Shapley computation. Unlike other near neighbor search methods, our method exhibits several merits: It is inexpensive in indexing, simple to implement, capable of providing a lower bound of the distance to the query point for unvisited points, and, last but not least, amenable to analysis as we will see in the subsequent sections.

The main idea is to pick an arbitrary point $z_{\text{mark}}$ as the *landmark* point, and sort all data points $D \cup D_{\text{test}}$ by their distances from $z_{\text{mark}}$ in ascending order. Denote by $r_{\text{mark}}(z)$ the rank of a point $z$ in the sorted list and by $B_i(z_{\text{test}}) \subseteq D$ the set of points in $D$ whose differences in rank from that of $z_{\text{test}}$ are within $i$, i.e.,

$$B_i(z_{\text{test}}) = \{z \in D \mid |r_{\text{mark}}(z) - r_{\text{mark}}(z_{\text{test}})| \leq i\}.$$

We call $i$ the *length* of the ball $B_i(z_{\text{test}})$. We also distinguish the left and right halves of the ball, i.e., $B_i^-(z_{\text{test}})$ and $B_i^+(z_{\text{test}})$, where

$$B_i^-(z_{\text{test}}) = \{z \in D \mid r_{\text{mark}}(z_{\text{test}}) - i \leq r_{\text{mark}}(z) < r_{\text{mark}}(z_{\text{test}})\},$$

and

$$B_i^+(z_{\text{test}}) = \{z \in D \mid r_{\text{mark}}(z_{\text{test}}) < r_{\text{mark}}(z) \leq r_{\text{mark}}(z_{\text{test}}) + i\}.$$

The motivation of using landmark points is that for any test point $z_{\text{test}}$, a data point $z$ around $z_{\text{test}}$ along the sorted list is likely to be its near neighbor. More importantly, it is possible to derive a lower bound of $\mathsf{d}(z, z_{\text{test}})$ by only considering the distances to the landmark point $z_{\text{mark}}$. That is, by triangle inequality,

$$\mathsf{d}(z, z_{\text{test}}) \geq |\mathsf{d}(z, z_{\text{mark}}) - \mathsf{d}(z_{\text{mark}}, z_{\text{test}})|. \tag{6}$$

This is particularly useful. For example, as the ball $B_i^-(z_{\text{test}})$ expands, the distance $\mathsf{d}(z, z_{\text{mark}})$ is non-increasing, and thus $\mathsf{d}(z, z_{\text{test}})$ is also non-decreasing, which gives us valuable information about $\mathsf{d}(z, z_{\text{test}})$ for any $z \in B_i^-(z_{\text{test}})$ even without actually visiting them. The case for $B_i^+(z_{\text{test}})$ is similar.

### 4.1   Optimized Landmark Selection

The tightness of the lower bound of $\mathsf{d}(z, z_{\text{test}})$ in Eq. (6) is highly dependent on the distance $\mathsf{d}(z_{\text{test}}, z_{\text{mark}})$. That is, the quality of our bounds can be significantly improved if the test points are close to the chosen landmark point. Therefore, we propose to strategically select multiple landmark points and assign each test point to its nearest landmark point.

More specifically, we propose to select $n_L$ landmark points $D_{\text{mark}} \subseteq D_{\text{test}}$ with a goal of minimizing the maximum distance of any test point to its nearest landmark point, i.e.,

$$\min_{D_{\text{mark}} \subseteq D_{\text{test}}} \max_{z_{\text{test}} \in D_{\text{test}}} \min_{z_{\text{mark}} \in D_{\text{mark}}} \mathsf{d}(z_{\text{test}}, z_{\text{mark}}).$$

The landmark points $D_{\text{mark}}$ defined above align exactly with our intention and naturally create a clustering structure through their associated regions of influence, where each region contains points closer to its landmark than to any other landmark. This is also known as the *Voronoi* partition induced by the landmark points. In subsequent sections, we will show that landmark-induced partitions can effectively reveal inherent structures in the data.

The above optimization problem turns out to be the well-studied metric $k$-center problem, which is known to be **NP**-hard. Worse still, this problem is impossible to approximate within a factor of 2, unless $\mathbf{P} = \mathbf{NP}$ [18].[2]

Fortunately, there exists a simple greedy algorithm that achieves the best possible approximation ratio of 2 in the worst case. This algorithm is called farthest-first traversal (FFT) [19], which as the name suggests, starts from an arbitrary point and iteratively selects the farthest point from the current set of landmark points until $n_L$ landmark points are selected. In other words, the next chosen landmark point $z$ maximizes the distance against the current set of landmark points, i.e.,

$$\mathsf{d}(z, D_{\mathrm{mark}}) = \min_{z_{\mathrm{mark}} \in D_{\mathrm{mark}}} \mathsf{d}(z, z_{\mathrm{mark}}).$$

A straightforward implementation of the FFT algorithm takes $\mathcal{O}(n_{\mathrm{test}} n_L^2 d)$ time, which may be too slow when $n_L$ is large.

To speed up the FFT algorithm, we first notice that $\mathsf{d}(z, D_{\mathrm{mark}})$ is non-increasing as $D_{\mathrm{mark}}$ grows. Thus, we can reduce a factor of $n_L$ in the running time by bookkeeping $\mathsf{d}(z, D_{\mathrm{mark}})$ for each point $z$ and updating it upon the selection of every new landmark point $z_{\mathrm{mark}}$. That is,

$$\mathsf{d}(z, D_{\mathrm{mark}}^{(i)}) = \min\{\mathsf{d}(z, z_{\mathrm{mark}}), \mathsf{d}(z, D_{\mathrm{mark}}^{(i)})\},$$

where $D_{\mathrm{mark}}^{(i)}$ is the set of landmark points selected before the $i$-th iteration and $D_{\mathrm{mark}}^{(i+1)} = D_{\mathrm{mark}}^{(i)} \cup \{z_{\mathrm{mark}}\}$. This avoids scanning the entire set of landmark points when computing $\mathsf{d}(z, D_{\mathrm{mark}})$ for each point $z$. Based on this observation, the running time is reduced to $\mathcal{O}(n_{\mathrm{test}} n_L d)$.

# 5 Fast Data Shapley Value Computation for Threshold-based KNN

As discussed previously, given $n_{\mathrm{test}}$ test points, the baseline approach in Section 3 computes the data Shapley values for all test points in $\mathcal{O}(d n_{\mathrm{test}} n + n_{\mathrm{test}} n \log n)$ time. In this section, we propose a fast algorithm to compute the data Shapley values for threshold-based KNN in provably less time.

Our main idea is to exploit the truncated structure of the threshold-based KNN classifier. We leverage the landmark-based near neighbor search introduced in Section 4 to shrink the search space of the near neighbors, and it turns out that the size of the search space for any test point can be effectively bounded when the dataset exhibits a stable clustering structure. In the remainder of this section, we first describe the proposed algorithm and then analyze its theoretical properties.

## 5.1 Algorithm Description

We have introduced the landmark-based near neighbor search in Section 4, and in this section, we show how to utilize it to compute data Shapley values efficiently.

First of all, we need to slightly adjust the FFT algorithm for our purpose. We denote by $C(z_{\mathrm{mark}})$ the cluster of points that are closer to $z_{\mathrm{mark}}$ than any other landmark point. The radius of $C(z_{\mathrm{mark}})$ is the maximum distance from $z_{\mathrm{mark}}$ to any point in $C(z_{\mathrm{mark}})$. In addition, we let $\tau_{D_{\mathrm{mark}}}$ be the maximum radius of all clusters induced by the landmark points in $D_{\mathrm{mark}}$. Note that $\tau_{D_{\mathrm{mark}}}$ is non-increasing as the number of landmark points increases. We make two adjustments to the original FFT algorithm. First, we run FFT over $D_{\mathrm{test}} \cup D$ instead of $D_{\mathrm{test}}$. Second, we require that $\tau_{D_{\mathrm{mark}}} \leq \tau$, which can be easily achieved by continuing the iterative process of FFT until the condition is met. This also means that it is not required to specify the number of landmark points in advance.

Since the threshold-based KNN classifier only considers data points within a distance of $\tau$ from each test point, an intuitive idea is to explore the search space provided by the landmark-based near neighbor search until we can certify that all remaining unvisited points are beyond a distance of $\tau$ from the test point. More specifically, we gradually expand the left half of the ball $B_i(z_{\mathrm{test}})$ centered at each test point $z_{\mathrm{test}}$ until the first point $z$ such that

$$|\mathsf{d}(z, z_{\mathrm{mark}}) - \mathsf{d}(z_{\mathrm{mark}}, z_{\mathrm{test}})| > \tau.$$

This immediately implies that the distance $\mathsf{d}(z, z_{\mathrm{test}})$ from any $z$ to $z_{\mathrm{test}}$ outside the left half is at least $\tau$. We also apply this process to the right half of the ball. Afterwards, we collect into $S$ all points in

---

[2]Recall that an algorithm is called $\gamma$-approximation if it returns solutions that in the worst case have cost no more than $\gamma$ times than the cost of the optimum solution.

---

**Algorithm 1:** Fast Data Shapley Value Computation for Threshold-based KNN

---
    **Input:** Integer $K$, radius $\tau$, datasets $D$ and $D_{\text{test}}$
    **Output:** Data Shapley values $\{s(z)\}_{z \in D}$
**1** Select landmark points $D_{\text{mark}}$ from $D \cup D_{\text{test}}$ by FFT such that $\tau_{D_{\text{mark}}} \leq \tau$;
**2** Assign each point in $D_{\text{test}}$ to its nearest point in $D_{\text{mark}}$;
**3 for** $z_{mark} \in D_{mark}$ **do**
**4**     Sort $D \cup D_{\text{test}}$ by their distances to $z_{\text{mark}}$ in ascending order;
**5** Initialize $s(z)$ with a default value of 0 for each $z \in D$;
**6 for** $z_{test} \in D_{test}$ **do**
**7**     Let $z_{\text{mark}}$ be the landmark point associated with $z_{\text{test}}$;
**8**     Expand the ball $B^-(z_{\text{test}})$ until the first point $z$ such that $\mathsf{d}(z_{\text{mark}}, z_{\text{test}}) - \mathsf{d}(z, z_{\text{mark}}) > \tau$;
**9**     Expand the ball $B^+(z_{\text{test}})$ until the first point $z$ such that $\mathsf{d}(z, z_{\text{mark}}) - \mathsf{d}(z_{\text{mark}}, z_{\text{test}}) > \tau$;
**10**     $S \leftarrow \{z \in B^-(z_{\text{test}}) \cup B^+(z_{\text{test}}) \mid \mathsf{d}(z, z_{\text{test}}) \leq \tau\}$;
**11**     Let $z_1, \ldots, z_{|S|}$ be the points in $S$ sorted by their distances to $z_{\text{test}}$ in ascending order;
**12**     $\phi_{z_{|S|}} \leftarrow \frac{K}{|S|} w(z_{|S|} \mid z_{\text{test}}) \mathbb{1}(y_{z_{|S|}} = y_{z_{\text{test}}})$;
**13**     **for** $i = |S| - 1, \ldots, 1$ **do**
**14**         $\phi_{z_i} \leftarrow \phi_{z_{i+1}} + \frac{\min(K, i)}{i} \left(w'_i - w'_{i+1}\right)$, where $w'_i = w(z_i \mid z_{\text{test}}) \mathbb{1}(y_{z_i} = y_{z_{\text{test}}})$;
**15**     $s(z) \leftarrow s(z) + \phi_z$ for each $z \in S$;
**16** $s(z) \leftarrow s(z)/n_{\text{test}}$ for each $z \in D$;
**17 return** $\{s(z)\}_{z \in D}$;

---

the ball that are within a distance of $\tau$ from $z_{\text{test}}$. Finally, we compute the data Shapley values for all the points in $S$ using the recursive formula in Theorem 2. The detailed procedure is described in Algorithm 1.

To show that Algorithm 1 can be provably faster than the baseline approach, we need to show that for any test point, the total number of points visited in the ball is strictly smaller than $n$. In the next subsection, we show that this is indeed the case under mild conditions.

## 5.2 Perturbation Resilience

Before we analyze the theoretical guarantee of Algorithm 1, we introduce a technical notion of perturbation resilience to help us precisely characterize the structure that exists in a dataset.

Worst-case analysis has been criticized for its over-pessimism and conservatism that fail to capture the real performance of many algorithms in practice. In particular, it ignores the structure that exists in real-world datasets. Therefore, in recent years, there is an active trend of *beyond worst-case analysis* in the literature to provide more realistic performance guarantees [20]. One notable example is the notion of *perturbation resilience* proposed by Bilu and Linial [21], which describes the stability of the clustering structure of a dataset under small perturbations.

We first define the notion of perturbation, which distorts the original distance function $\mathsf{d}(\cdot, \cdot)$ by a factor of at most $\xi \geq 1$.

**Definition 1** (Perturbation). *Given a clustering instance $(D, \mathsf{d})$, a $\xi$-perturbation of $\mathsf{d}$ is a new distance function $\mathsf{d}'$ such that $\mathsf{d}(x, y) \leq \mathsf{d}'(x, y) \leq \xi \mathsf{d}(x, y)$ for all $x, y \in D$.*

Note that the $\xi$-perturbation of $\mathsf{d}$ may no longer be a metric distance. As $\xi$ increases, a larger perturbation is allowed. Then, we say that a clustering instance is perturbation resilient if its optimal clustering remains unchanged up to such a small perturbation of $\mathsf{d}$.

**Definition 2** (Perturbation Resilience (PR)). *A clustering instance $(D, \mathsf{d})$ is said to be $\xi$-perturbation resilient if the optimal clustering remains unchanged up to a $\xi$-perturbation of $\mathsf{d}$.*

Here, the optimal clustering depends on the specific clustering objective. For example, in the metric $k$-center clustering, the optimal clustering is the one that minimizes the maximum cluster radius for a specific number of clusters $k$.

Intuitively, a clustering instance is $\xi$-perturbation resilient if the optimal clustering is stable. Conversely, if the optimal clustering can easily change under a small perturbation, then it is less meaningful to study the clustering structure of the dataset in the first place. Note that perturbation resilience does not mean that the clustering will necessarily become easy. In fact, as indicated in [22], no polynomial-time algorithm can solve the metric $k$-center clustering problem for $\xi$-perturbation resilient instances with any $\xi < 2$ unless $\mathbf{RP} = \mathbf{NP}$. Although the worst-case approximation ratio of FFT for metric $k$-center clustering is at least 2, it turns out that it can recover the optimal clusters in a clustering instance if it is 2-perturbation resilient.

**Theorem 3** (Balcan et al. [22]). *Let $(D, \mathsf{d})$ be a clustering instance. If $(D, \mathsf{d})$ is 2-perturbation resilient, then* FFT *recovers the optimal clusters of $(D, \mathsf{d})$.*

Furthermore, Theorem 3 can be extended to show that any $\gamma$-approximation solution can optimally recover the clusters under any $\gamma$-perturbation of d. Theorem 3 opens up a new avenue for analyzing the performance of our landmark-based near neighbor search, and in turn, of Algorithm 1.

### 5.3 Theoretical Analysis

In this subsection, we show that Algorithm 1 is provably faster than the baseline approach if the dataset exhibits perturbation resilience. We first point out that our landmark-based near neighbor search can be seen as a soft version of the $k$-center clustering. Then, we show that perturbation resilience provides sufficient separation between clusters to restrict the computation of the data Shapley values within each cluster, which leads to a provable speedup.

If we can treat landmark points as the selected centers of the $k$-center clustering, then they virtually partition the dataset into $n_L$ clusters. Given a landmark point $z_{\mathrm{mark}}$ and the sorted list of $D \cup D_{\mathrm{test}}$ by their distances to $z_{\mathrm{mark}}$ in ascending order, we hope that the points from its cluster $C(z_{\mathrm{mark}})$ are ranked before the points in other clusters.

The next challenge is how to ensure that every point that is within a distance of $\tau$ from $z_{\mathrm{test}}$ is in the same cluster as $z_{\mathrm{test}}$. Note that this is non-trivial because $z_{\mathrm{test}}$ may not be a landmark point (i.e., a center), and there is no guarantee that any two arbitrary points with a distance less than $\tau$ will be in the same cluster, no matter what size the cluster radius is. For example, consider two clusters that overlap with each other and two points that lie in the overlapping region. This is crucial for the algorithm to restrict the computation of every test point within the cluster to which it belongs.

We discover that perturbation resilience provides sufficient separation between clusters to address the above challenges. Formally, we prove the following theorem.

**Theorem 4.** *Let $(D \cup D_{test}, \mathsf{d})$ be a clustering instance with $n_{test} = \mathcal{O}(n)$. If it is 3-perturbation resilient for metric $k$-center with respect to a cluster number $k^*$ and a maximum cluster radius $\tau^* \geq \tau$, then Algorithm 1 with $n_L = k^*$ returns the exact data Shapley values of each point in $D$ in $\mathcal{O}(n_L n (d + \log n) + n_{test} s (d + \log s))$ time, where $s$ is the size of the largest optimal cluster.*

See proof in Appendix A. When $k^* < n_L$, we will show in Appendix A that the same time complexity as Theorem 4 still holds, albeit with a slightly larger 4-perturbation. The above results show that Algorithm 1 provides a provable speed-up for a wide range of numbers of clusters $n_L$. For example, let $n_{\mathrm{test}} = \Omega(n)$, and we can expect a running time of $\mathcal{O}(n^{1.5}(d + \log n))$ with $n_L = \sqrt{n}$ when the sizes of the clusters are comparable. This improves over the previous $\mathcal{O}(n^2(d + \log n))$ time.

## 6 Experiments

In this section, we evaluate the performance of the proposed methods on synthetic and real-world datasets. We aim to answer the following two research questions: (1) What are the effects of the parameters and design choices on the performance of the fast algorithm for threshold-based KNN (Algorithm 1)? (Section 6.1) (2) How much faster do the proposed methods compute the data Shapley values compared to the baseline approach in Section 3? (Section 6.2) Our source code is published for reproducibility.[3]

**Datasets.** We used both synthetic and real-world datasets in the experiments. The former allows us to experiment freely with a wide range of data characteristics. We select a collection of real-world

---

[3]`https://github.com/Guangyi-Zhang/tknn-data-shapley`

datasets as listed in Table A1. The dataset size $|D|$ ranges from 10 K to 1 M, and we set $|D_{\text{test}}|$ to be 0.2%-1% of $|D|$. Thus, the total size of $|D| \cdot |D_{\text{test}}|$ is up to the order of $10^{10}$.

**Experimental Environment.** All algorithms were implemented in Python 3.11. All experiments were carried out on a Linux server equipped with 64 CPUs of Intel(R) Xeon(R) Platinum 8358P CPU @ 2.60 GHz and 1511 GB RAM.

## 6.1 Effects of Parameters and Design Choices

In this subsection, we investigate the effects of the parameters and design choices on the performance of Algorithm 1. The default values for the parameters are $K = 5$, $n_L = 50$, $\tau/d = 0.2$, $\sigma = 0.1$, and FFT for landmark selection. Note that since the data features are normalized to be in the range of $[0, 1]$, the value of $\tau/d$ measures the maximum proportion of features of a data point that can deviate arbitrarily from a test point, while the point remains to be considered as a neighbor in threshold-based KNN models.

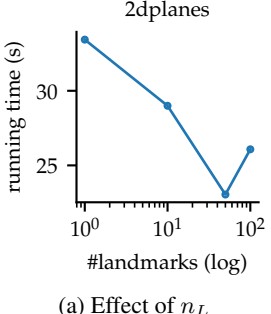
(a) Effect of $n_L$

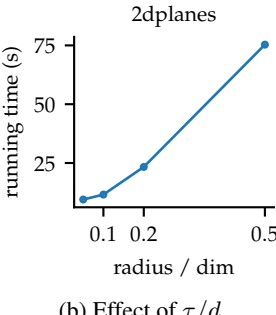
(b) Effect of $\tau/d$

Figure 2: Effect of parameters $n_L$ and $\tau/d$ on the performance of Algorithm 1.

**Effect of the Number of Landmark Points $n_L$.** We vary the number of landmark points $n_L$ from 1 to 100, and plot the corresponding running time on the 2dplanes dataset in Fig. 2a. As shown, the running time of Fig. 2a first decreases and then increases when more landmark points are selected. This is because every test point can be assigned to a landmark point that is closer to it, and the total number of data points encountered during the ball expansion for a test point is effectively reduced when there are more landmark points. However, the running time increases later when this benefit is outweighed by the high overhead of preprocessing the landmark points. Recall that it is necessary to sort all the data points in $D \cup D_{\text{test}}$ for each landmark point.

**Effect of the Ratio of Radius $\tau$ and Dimension $d$.** We vary the value of $\tau/d$ from 0 to 0.5, and plot the corresponding running time on the 2dplanes dataset in Fig. 2b. It is expected that the running time increases as the radius $\tau$ grows, because a larger $\tau$ allows more data points to be considered as neighbors of a test point. But fortunately, a small $\tau$ is enough in practice due to the localized nature of KNN models.

## 6.2 Performance Comparison

In this subsection, we compare the performance of the proposed methods with the baseline approach in Section 3 and the state-of-the-art threshold-based approach TNN by Wang et al. [14]. Note that the baseline approach has to sort all the data points in $D$ for each test point, regardless of the radius value $\tau$. We report the average running time and standard deviation over three runs in Table 1, where the fastest running time is highlighted in bold. We also compare their performance for a popular downstream task of mislabel detection; see Appendix C.1 for more details. The default values for the parameters are $K = 5$, $n_L = 50$, and FFT for landmark selection.

Based on the results in Table 1, we observe that our proposed methods consistently outperform the baseline approach across all datasets. The speedup is substantial, with our fastest method running up to 25× faster than the baseline and 11× faster than TNN. The performance advantage of our methods becomes more pronounced as the dataset size increases.

Table 1: Running time (in seconds) comparison of different methods.

| Dataset | Baseline | Algorithm 1 | | | TNN |
|---|---|---|---|---|---|
| | N/A | $\tau/d = 0.05$ | $\tau/d = 0.1$ | $\tau/d = 0.2$ | $\tau/d = 0.05$ |
| magic | $7.05 \pm 6.48$ | $2.68 \pm 0.02$ | $3.17 \pm 0.05$ | $4.70 \pm 0.21$ | $\mathbf{1.29} \pm 0.06$ |
| 2dplanes | $85.83 \pm 3.00$ | $\mathbf{8.44} \pm 0.15$ | $11.59 \pm 0.24$ | $26.75 \pm 0.20$ | $27.28 \pm 25.13$ |
| cifar10 | $148.60 \pm 0.61$ | $75.83 \pm 0.50$ | $86.57 \pm 2.35$ | $144.05 \pm 1.67$ | $\mathbf{48.16} \pm 44.94$ |
| dota2 | $445.44 \pm 4.89$ | $249.36 \pm 1.74$ | $267.76 \pm 7.24$ | $436.65 \pm 6.32$ | $\mathbf{142.52} \pm 132.03$ |
| skin | $2035.00 \pm 142.90$ | $\mathbf{156.88} \pm 1.90$ | $159.37 \pm 126.42$ | $243.79 \pm 207.22$ | $228.59 \pm 2.52$ |
| covtype | $10324.36 \pm 131.70$ | $\mathbf{1918.74} \pm 16.94$ | $4135.27 \pm 27.07$ | $9107.61 \pm 323.44$ | $5009.56 \pm 10.33$ |
| emnist | $6090.47 \pm 19.04$ | $\mathbf{2508.41} \pm 73.36$ | $3558.89 \pm 130.26$ | $5988.33 \pm 384.80$ | $3221.94 \pm 7.48$ |
| poker | $12289.43 \pm 41.26$ | $\mathbf{488.95} \pm 4.73$ | $983.97 \pm 0.91$ | $3012.10 \pm 46.34$ | $5542.69 \pm 23.88$ |

Algorithm 1 excels on lower-dimensional datasets (see a controlled experiment in Appendix C.2), such as 2dplanes and poker, where it achieves the fastest runtime when the radius is small. As expected, the performance of Algorithm 1 degrades as the radius increases. However, the running time never exceeds that of the baseline approach, even when $\tau/d$ is as large as 0.2. Actually, its worst-case running time is about the same as that of the baseline approach, with the negligible overhead of preprocessing a few landmark points. When the dataset shows well-defined clusters, its running time can be provably better. These results confirm that the proposed algorithm significantly accelerates the computation of data Shapley values, while retaining a robust worst-case running time that is at least as fast as the baseline approach.

## 7 Conclusion

In this paper, we addressed the challenge of efficiently computing data Shapley values for nearest neighbor algorithms. We leveraged the distance-based local structure in the data space to decompose the global valuation problem into smaller, localized computations. For threshold-based KNN classification, we proposed an algorithm with provable speedups under mild assumptions compared to existing methods. Our comprehensive empirical evaluation on synthetic and real-world datasets verifies the significant speed-ups offered by our proposed methods.

We acknowledge several limitations of our work. The perturbation resilience condition is hard to verify in practice. The landmark-based near neighbor search may suffer from the curse of dimensionality. Potential future research directions include extending our approach to regression tasks, exploring alternative utility functions, and investigating fast algorithms for more general KNN models.

## Acknowledgments and Disclosure of Funding

G. Zhang was supported by the Guangdong Provincial College Youth Innovative Talent Project (Grant No. 2025KQNCX075), Natural Science Foundation of Top Talent of SZTU (Grant No. GDRC202520), SZTU University Research Project (No. 20251061020002). Y. Wang was supported by the National Natural Science Foundation of China (Grant No. 62202169). C. Chai was supported by the NSF of China (62472031), the National Key Research and Development Program of China (2024YFC3308200), Beijing Nova Program, CCF-Baidu Open Fund (CCF-Baidu202402). Q. Liu was supported by fundamental research funds for the central universities of Ministry of Education of China (SWU-KR24043). W. Wang was supported by Guangdong Provincial Key Lab of Integrated Communication, Sensing and Computation for Ubiquitous Internet of Things (No.2023B1212010007, SL2023A03J00934), Guangzhou Municipal Science and Technology Project (No. 2023A03J0003, 2023A03J0013 and 2024A03J0621).

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

# A Missing Proofs

**Theorem 2.** *Fixing a test point, for any $i < n$, we have*

$$s_i = s_{i+1} + \min(K, i)\frac{w_i\mathbb{1}(y_i = y_{test}) - w_{i+1}\mathbb{1}(y_{i+1} = y_{test})}{i} \quad and \quad s_n = \frac{K}{n}w_n\mathbb{1}(y_n = y_{test}).$$

*Proof of Theorem 2.* For $s_n$, i.e., the Shapley value of the farthest data point, it is easy to see that

$$s_n = \frac{1}{n}\sum_{k=0}^{K-1}\frac{1}{\binom{n-1}{k}}\sum_{S \subseteq D - z_n, |S| = k} v(S + z_n) - v(S)$$

$$= \frac{1}{n}\sum_{k=0}^{K-1}\frac{1}{\binom{n-1}{k}}\sum_{S \subseteq D - z_n, |S| = k} w_n\mathbb{1}(y_n = y_{\text{test}})$$

$$= \frac{K}{n}w_n\mathbb{1}(y_n = y_{\text{test}}). \tag{7}$$

Since for any $S$ such that $|S| \geq k$, $z_n$ will not make it into the top-$k$ and thus $v(S + z_n) - v(S) = 0$.

Suppose that we have already computed $s_{i+1}, \ldots, s_n$, and now we want to compute $s_i$. By Lemma 1, we only need to pay attention to $S \subseteq D - z_i - z_{i+1}$. Divide $S$ into two parts: $S_1 = S \cap \{z_1, \ldots, z_{i-1}\}$ and $S_2 = S \cap \{z_{i+2}, \ldots, z_n\}$. Notice that if $|S_1| \geq K$, $v(S + z_i) = v(S + z_{i+1}) = v(S)$. Therefore, we only need to consider the case when $|S_1| < K$. We have

$$s_i - s_{i+1} = \frac{1}{n-1}\sum_{k=0}^{n-2}\frac{1}{\binom{n-2}{k}}\sum_{|S_1 \cup S_2| = k, |S_1| < K} v(S + z_i) - v(S + z_{i+1})$$

$$= \frac{1}{n-1}\sum_{k=0}^{n-2}\frac{1}{\binom{n-2}{k}}\sum_{|S_1 \cup S_2| = k, |S_1| < K} w'_i - w'_{i+1}$$

$$= \frac{w'_i - w'_{i+1}}{n-1}\sum_{k=0}^{n-2}\frac{1}{\binom{n-2}{k}}\sum_{k_1=0}^{\min(K-1,k)}\binom{i-1}{k_1}\binom{n-i-1}{k-k_1}$$

$$= \frac{w'_i - w'_{i+1}}{n-1}\frac{\min(K,i)(n-1)}{i}$$

$$= \frac{\min(K,i)}{i}(w'_i - w'_{i+1}),$$

where $w'_i = w_i\mathbb{1}(y_i = y_{\text{test}})$. See [11] for more details on the identity in the second to last step. □

**Theorem 4.** *Let $(D \cup D_{test}, d)$ be a clustering instance with $n_{test} = \mathcal{O}(n)$. If it is 3-perturbation resilient for metric $k$-center with respect to a cluster number $k^*$ and a maximum cluster radius $\tau^* \geq \tau$, then Algorithm 1 with $n_L = k^*$ returns the exact data Shapley values of each point in $D$ in $\mathcal{O}(n_L n(d + \log n) + n_{test}s(d + \log s))$ time, where $s$ is the size of the largest optimal cluster.*

*Proof of Theorem 4.* We first prove that 3-perturbation resilience guarantees a useful technical lemma.

**Lemma 5.** *Given a center $z_{mark}$ and any point $z$ from a different cluster than that of $z_{mark}$, we have $d(z, z_{mark}) > 2\tau^*$.*

*Proof.* Assume for contradiction that $d(z, z_{\text{mark}}) \leq 2\tau^*$. We can create a 3-perturbation of d such that a set of centers $D_{\text{mark}} - z_{\text{mark}} + z$ yield a 3-approximation of the $k$-center clustering. This contradicts the assumption since any 3-approximation solution recovers the optimal clustering, following Theorem 3.

The 3-perturbation is given as follows.

$$d'(x, y) = \begin{cases} \min\{3\tau^*, 3d(x, y)\} & \text{if } x = z \text{ and } y \in C(z_{\text{mark}}), \\ 3d(x, y) & \text{otherwise.} \end{cases}$$

It is easy to verify that $\mathsf{d}'$ is a 3-perturbation of $\mathsf{d}$. Besides, assigning every point $x \in C(z_{\text{mark}})$ to the new center $z$ respects the inequality

$$\mathsf{d}(x, z) \le \mathsf{d}(x, z_{\text{mark}}) + \mathsf{d}(z_{\text{mark}}, z) \le 3\tau^*.$$

The assignments for other data points are either intact or better. Thus, the new centers indeed yield a 3-approximation, completing the proof. $\square$

We also need the help from another lemma.

**Lemma 6** (Chekuri and Gupta [23]). *Under 2-perturbation resilience, for any two different optimal clusters $C_i, C_j$, we have*

$$\mathsf{d}(x, y) < \mathsf{d}(x, z) \quad \text{and} \quad \tau^* < \mathsf{d}(x, z)$$

*for any $x, y \in C_i$ and $z \in C_j$.*

As a result, when $k^* = n_L$, the clusters induced by $D_{\text{mark}}$ recover the optimal clusters by Theorem 3. This implies that the points assigned to the optimal cluster of $z_{\text{mark}}$ will be ranked before the points in other clusters, following Lemma 6. What is more, by Lemma 5, the subset $S$ for each test point $z_{\text{test}}$ associated with $z_{\text{mark}}$ in Algorithm 1 will collect only points from the optimal cluster $z_{\text{test}}$ belongs to, because $|\mathsf{d}(z, z_{\text{mark}}) - \mathsf{d}(z_{\text{mark}}, z_{\text{test}})| \ge \tau^* \ge \tau$ for any $z$ from a different cluster. This verifies the stated time complexity.

What is left to show is the correctness, i.e., every point that is within a distance of $\tau$ from $z_{\text{test}}$ must stay in the same optimal cluster as $z_{\text{test}}$, which directly follows from Lemma 6 and $\tau^* \ge \tau$, and they will be collected into the subset $S$ as $|\mathsf{d}(z, z_{\text{mark}}) - \mathsf{d}(z_{\text{mark}}, z_{\text{test}})|$ is a lower bound of $\mathsf{d}(z, z_{\text{test}})$. This completes the proof. $\square$

When $k^* < n_L$, we can show that the same time complexity in Theorem 4 still holds, albeit with a slightly larger 4-perturbation.

**Theorem 7.** *Let $(D \cup D_{test}, \mathsf{d})$ be a clustering instance with $n_{test} = \mathcal{O}(n)$. If $(D \cup D_{test}, \mathsf{d})$ is 4-perturbation resilient for metric $k$-center with respect to a cluster number $k^*$ and a maximum cluster radius $\tau^* \ge \tau$, then Algorithm 1 returns the exact data Shapley values of each data point in $D$ in time $\mathcal{O}(n_L n(d + \log n) + n_{test} s(d + \log s))$, where $s$ is the size of the largest optimal cluster.*

*Proof.* We first show a stronger lemma than Lemma 5 under 4-perturbation. We omit the proof since it is similar to that of Lemma 5.

**Lemma 8.** *Given a optimal center $c$ and any point $z$ from a different cluster than that of $c$, we have $\mathsf{d}(z, c) > 3\tau^*$.*

Lemma 8 implies a stronger variant of Lemma 6.

**Lemma 9.** *If $(D, \mathsf{d})$ is 2-perturbation resilient, then for any two different optimal clusters $C_i, C_j$, we have $2\tau^* < \mathsf{d}(x, z)$ for any $x \in C_i$ and $z \in C_j$.*

*Proof.* Let $c$ be the optimal center of $C_j$. By triangle inequality, we have

$$\mathsf{d}(x, z) \ge |\mathsf{d}(x, c) - \mathsf{d}(c, z)| > 2\tau^*.$$

$\square$

As a result of Lemma 9, we can relax the requirement that every landmark point has to correspond to an optimal center, and let it be any point instead. The statement can be proved by similar arguments as in the proof of Theorem 4. $\square$

# B Expanded Related Work

## B.1 Shapley Values

Shapley values [6] originated in cooperative game theory as a method for fairly distributing gains among players based on their marginal contributions and have been widely adopted in various domains such as economics [24]. In machine learning, the most well-known application is attribution methods such as SHAP [9], where Shapley values are computed for each feature of a given data point as a form of feature importance scores. Recently, Shapley values have been adapted to quantify the contribution of individual training examples to model performance [7]. Computing exact Shapley values is well-known to be expensive and has been shown to be $\#\mathbf{P}$-hard in certain games [10]. Such a computational challenge has motivated various approximation techniques, including mostly Monte Carlo sampling [25–27] and specialized algorithms for specific games [28]. Our work falls into the latter category.

## B.2 Data Valuation

Data valuation aims to assign importance scores to training examples, with the hope of identifying valuable or harmful data points [2–4]. The dominant approaches are based on the concept of leave-one-out (LOO), which measures the marginal contribution of a data point to the utility function when it is removed from the training procedure. In classification settings, a common choice for the utility function is the test accuracy of a model trained on the input. Data Shapley [7] and its variants such as Beta Shapley [29], Data Banzhaf [30], and least core [31], are all based on the LOO principle, but differ in the way marginal contributions are aggregated.

Beyond Shapley values, there exist other approaches, and we discuss some notable ones below. Feldman and Zhang [32] simulate the data values by LOO retraining albeit constrained on a small sample of training data, while DataModels [33] sacrifice the exactness of LOO to achieve better scalability by model predictions. Another line of popular methods are gradient-based. TracIn [34] estimates the importance of a training example by tracing the change in test loss caused by the example during the training process. Variations of influence functions [35, 36] have their roots in robust statistics [37] and offer a gradient-based approximation of LOO values.

## B.3 KNN Shapley Values

The KNN model provides a unique opportunity for efficient computation of data Shapley values. Jia et al. [11] are the first to discover an efficient algorithm for computing unweighted KNN Shapley values with a complexity of $\mathcal{O}(dn_{\text{test}}n + n_{\text{test}}n \log n)$. This is a significant improvement over general Shapley computation methods, making it feasible for datasets of moderate size. Wang and Jia [13] provide refinements to the unweighted KNN utility function. Building on this foundation, Wang et al. [15] tackle the weighted KNN case, which turns out to be more challenging due to the normalization factor in the utility function. They propose a dynamic programming algorithm for a hard-label weighted KNN utility function. Furthermore, Wang et al. [14] addressed privacy concerns in computing KNN Shapley values and offered formal privacy guarantees for a threshold-based KNN utility function. Note that in their utility function, all near neighbors within the ball around a test point are equal. Our work focuses on accelerating the computation of KNN Shapley values.

## B.4 Clustering and Perturbation Resilience

Clustering is a common technique to exploit the structure of a dataset. Among many clustering methods, the $k$-center clustering is one of the most popular. It is well-known that the $k$-center clustering problem is $\mathbf{NP}$-hard, and FFT is proved to be 2-approximate in the worst case. To overcome the over-pessimism and conservatism of worst-case analysis, in recent years, *beyond worst-case analysis* (BWCA) has received increasing attention. One popular BWCA approach for clustering problems is to define a notion of stability, and *perturbation resilience* is one classic stability measure [21]. It has been shown that multiple clustering problems admit polynomial-time algorithms under some degree of perturbation resilience [38, 39]. Balcan et al. [22] analyze the FFT algorithm as a robust solution in BWCA scenarios. Chekuri and Gupta [23] study $k$-center clustering in the presence of outliers. Our work leverages these existing understandings about perturbation resilience to provably accelerate the computation of KNN Shapley values.

Table A1: Statistics of datasets used in the experiments.

| Dataset | $|D|$ | $|D_{\text{test}}|$ | $d$ | $|D| \cdot |D_{\text{test}}| \cdot d$ |
|---------|-------|---------------------|-----|--------------------------------------|
| magic | 15 063 | 153 | 10 | 23 046 390 |
| 2dplanes | 40 360 | 408 | 10 | 164 668 800 |
| cifar10 | 49 500 | 500 | 512 | 12 672 000 000 |
| dota2 | 91 722 | 927 | 125 | 10 628 286 750 |
| skin | 194 084 | 1961 | 3 | 1 141 796 172 |
| covtype | 578 106 | 2906 | 54 | 90 718 705 944 |
| emnist | 696 536 | 1396 | 512 | 497 850 499 072 |
| poker | 998 000 | 2000 | 10 | 19 960 000 000 |

## C  Additional Experimental Results

### C.1  Mislabel Detection

We adopt the popular downstream task of mislabel detection, where 5% of the training points are randomly mislabeled, and we try to detect them using the data points with the lowest data values. We use the F1 score to evaluate the performance of valuation methods. The F1 score of our method is slightly worse than that of the un-thresholded baseline (no more than 1% worse), which is unsurprising given their similar formulations. The F1 score of TNN is 20-50% worse than the others in most datasets. This is most likely due to the fact that TNN does not utilize the ranking signal among neighborhood points. We tune the radius on a validation set sampled from the noisy training set. We use the implementation of TNN by its authors [14].

### C.2  Curse of Dimensionality

We conduct a controlled experiment to isolate the "curse of dimensionality" effect. We perform projections by a Gaussian random matrix on the cifar10 dataset to reduce its dimensionality to 10, and use the same ratio of $\tau/d$ as before. The baseline takes 116s (148s previously) and our method with $\tau/d = 0.05$ takes 13s (75s previously). The running time of all methods decreases, due to the smaller overhead for distance computation, but the ratio of the running time of the baseline to ours increases. Therefore, it is indeed helpful to reduce the dimensionality by random projections.

