# OpenReview forum: "Localized Data Shapley: Accelerating Valuation for Nearest Neighbor Algorithms"
_NeurIPS.cc/2025/Conference — NeurIPS 2025 poster_

### Official Review · Reviewer_JeLY · 2025-07-01

**Clarity:** 3
**Significance:** 2
**Originality:** 3
**Rating:** 4
**Confidence:** 2

**Summary:**

This paper introduces a localized data Shapley framework to accelerate the valuation of data points for K-nearest neighbor (KNN) models. The authors leverage distance-based local structure to decompose the global valuation problem into smaller computations. They propose an efficient algorithm for threshold-based KNN that achieves provable speedups under perturbation resilience assumptions, with experiments showing up to 25x speedup compared to baseline methods.

**Questions:**

- Could you provide some practical guidance on estimating $k^\*$ or choosing $n_L$ for a new dataset where the clustering structure is unknown?
- In line 217, you mention that $\tau_{D_{\textrm{mark}}} \le \tau$ is "easily achieved by continuing the iterative process of FFT." How many landmarks does this typically require in practice? Does this ever result in $n_L > k^\*$?

**Ethical Concerns:**

["NO or VERY MINOR ethics concerns only"]

**Final Justification:**

The author answered my questions while my concerns about weakness were still valid. Thus, I will not raise my score and keep my original rating.

**Limitations:**

See weaknesses above.

**Quality:**

3

**Strengths And Weaknesses:**

### Strengths
- Clear presentation and solid theoretical foundation. The paper is really well-written with a logical flow from preliminaries (Section 2) through the baseline approach (Section 3) to the proposed methods. The mathematical notation is consistent, and the recursive formula in Theorem 2 provides a nice foundation for understanding how Shapley values can be computed efficiently for KNN models.
- Practical algorithm design with theoretical guarantees. The landmark selection using farthest-first traversal (FFT) is a smart choice that balances theoretical guarantees with practical efficiency. The algorithm gracefully handles the worst case.
- Novel use of perturbation resilience for algorithm analysis. The connection between perturbation resilience (Definition 2) and the efficiency of landmark-based near neighbor search is creative. The authors cleverly show that 3-perturbation resilient instances allow Algorithm 1 to restrict computations within clusters, leading to the complexity in Theorem 4.
- Impressive empirical performance gains. The experimental results in Table 1 are quite compelling - the proposed Algorithm 1 achieves speedups up to 25x across different datasets.

### Weaknesses
- Limited discussion on determining $k^\*$. While the paper assumes knowledge of the optimal cluster number $k^\*$ for the theoretical analysis, there is insufficient guidance on how to determine or estimate this value in practice. The authors mention that "perturbation resilience condition is hard to verify in practice" (line 353), but don't provide any heuristics or methods to estimate $k^\*$ for real datasets, which limits the practical applicability of the theoretical guarantees.
- Missing ablation study on $n_L$ in final experiments. Section 6.1 shows the effect of $n_L$ on the 2dplanes dataset (Figure 1a), but Section 6.2 only reports results with a fixed $n_L = 50$ for all datasets. Given that the theoretical speedup depends heavily on $n_L$ and the paper claims "significant speedup even when $n_L$ is as large as $\sqrt{n}$" (line 293), it would strengthen the paper to show how performance varies with different $n_L$ values across all datasets in Table 1.
- Restrictive threshold-based KNN variant. The main algorithm focuses on threshold-based KNN (TKNN), which has "additional desirable privacy-friendly properties" (line 114), but this variant is less commonly used than standard KNN. While the authors claim TKNN is "more robust" (line 112), they don't provide empirical evidence comparing the actual prediction performance of TKNN vs standard KNN, making it unclear whether users would want to adopt this variant just for faster Shapley computation.

---

> ### Author Rebuttal · Authors · 2025-07-28
>
> We thank the reviewer for the time and effort in reviewing our paper.
> We answer your questions below.
>
> **Q1**:
> Could you provide some practical guidance on estimating $k^*$ or choosing $n_L$ for a new dataset where the clustering structure is unknown?
>
> **A1**:
> Deciding the number of landmarks $n_L$, or in general, the number of clusters $k^*$ for geometric clustering, is a long-standing challenging problem.
> What we did in Section 6.1 is essentially the most popular heuristic for this problem, which is also known as the Elbow method.
> That is, to test the metric of interest (e.g., the clustering error or running time in our context) with varying $n_L$, and choose the $n_L$ that gives the best performance.
>
>
> **Q2**:
> In line 217, you mention that $\tau_{D_{mark}} \le \tau$ is "easily achieved by continuing the iterative process of FFT." How many landmarks does this typically require in practice? Does this ever result in $n_L > k^*$?
>
> **A2**:
> Unfortunately, it is unlikely that we can find the optimal value of $k^*$ for general datasets.
> We do test the number of landmarks $n_L$ by running the FFT algorithm on the datasets in our experiments.
> We found that typically it takes up to hundreds of landmarks to achieve $\tau_{D_{mark}} \le \tau$ for a small $\tau$.
> This may result in a larger overhead in practice.
> Thus, in practice, we recommend using the Elbow method (see Figure 1) to fix some small number of landmarks.

---

> > ### Comment · Reviewer_JeLY · 2025-08-04
> >
> > Thank you for your response. I will keep my positive score.

---

### Official Review · Reviewer_WTuj · 2025-07-02

**Clarity:** 3
**Significance:** 3
**Originality:** 4
**Rating:** 4
**Confidence:** 4

**Summary:**

The paper presents a new technique to calculate the Data Shapley values for KNN models. The proposed method addresses the significant computational cost of existing KNN-Shapley when applied to large test sets. The evaluation shows some promising efficient improvements.

**Questions:**

1. Compare with TKNN-Shapley in terms of compute efficiency.
2. Evaluate the utility of proposed data values on popular downstream tasks.
3. Discuss the online test query setting.

I'm happy to raise my scores if the weaknesses are addressed.

**Ethical Concerns:**

["NO or VERY MINOR ethics concerns only"]

**Final Justification:**

The technique proposed is a novel addition to the line of work on the Shapley value for KNN models. During the rebuttal, the authors have addressed the concerns regarding the comparison with literature and potential extension to online settings. Given the current shape, the paper is a solid contribution.

**Limitations:**

Yes

**Quality:**

3

**Strengths And Weaknesses:**

Strengths:

1. The technique proposed is novel, and appears to be an interesting addition to the line of work on the Shapley value for KNN models.
2. The evaluation shows promising efficiency gains.

Weaknesses:
1. It is important to point out that the utility function underlying the Shapley value that the paper is focused on is a different utility function from the original KNN-Shapley value. Specifically, the paper considers the utility for some distance-weighted threshold-based KNN. The original paper of KNN-Shapely just considered the standard unweighted KNN. If one is allowed to change the utility function, then Wang et al. [1] considers the unweighted threshold-based KNN and the computation complexity of this method is only linear in both training and test size, getting rid of the log N factor. The paper should compare the computational efficiency with this method.
2. On a related note, since the paper changes the utility function, it is crucial to examine whether the resulting Shapely value can still detect the data quality differences as effectively as KNN-Shapley and TKNN-Shapley [1].
[1] Wang et al. Threshold KNN-Shapley: A Linear-Time and Privacy-Friendly Approach to Data Valuation
3. The proposed algorithm requires to sort all data points in training and test sets  for each landmark point, which results in a complexity of O((N_train+N_test)log(N_train+N_test) N_landmark). If the test points are coming on the fly, this computation cost would be problematic as we need to re-do the sorting whenever a new test point arrives. Yet KNN-Shapley does not have this landmark building setup and circumvents the costs. The paper lacks some discussion on how this landmark building costs would affect the overall efficiency in different contexts.

---

> ### Author Rebuttal · Authors · 2025-07-28
>
> We thank the reviewer for the time and effort in reviewing our paper.
> We answer your questions below.
>
> **Q1**:
> Compare with TKNN-Shapley in terms of compute efficiency, and evaluate the utility of proposed data values on popular downstream tasks.
>
> **A1**:
> First of all, we believe it is more suitable to call this method "TNN-Shapley" instead of "TKNN-Shapley", which highlights that fact that it does not require a parameter $K$.
> Basically, TNN does not perform any ranking over the neighbors of the test point, which is why it only cost $O(n)$ time instead of $O(n \log n)$ time.
> It fetches all neighbors that are within the radius $\tau$ (in $O(n)$ time) and derives a value from the counts of these neighbors in each class.
> Thus, it should not be surprising that it tends to result in worse accuracy than methods with KNN formulations (see below).
>
> We adopt the popular downstream task of mislabel detection, where 5% of the training points are randomly mislabeled, and we try to detect them using the data points with the lowest data values.
> We use the F1 score to evaluate the performance of valuation methods.
> The F1 score of our method is slightly worse than that of the
> un-thresholded kNN-Shapley (no more than 1% worse).
> The F1 score of TNN is 20-50% worse than the others in 9 out of 11 datasets.
> That is, if our F1 score is 70%, TNN's F1 score is at most 50%.
> We tune the radius $\tau$ on a validation set sampled from the noisy training set.
> We use the implementation of TNN by its authors.
>
> In terms of efficiency, TNN is indeed faster than the baseline KNN-Shapley.
> The complexity of TNN is $O(n_{test} \cdot n_{train})$ instead of $O(n_{test} \cdot n_{train} \log n_{train})$, but still quadratic if the number of test points is large.
> Thus, it is slower than our method, especially over large datasets.
> For example, its running time is 5542.69 ($\pm$ 23.88) seconds for the Poker dataset, which is about 11$\times$ slower than ours.
>
> We will add this comparison to the revised paper.
>
>
>
> **Q2**:
> Discuss the online test query setting.
>
> **A2**:
> Extending the technique to the online test query setting is an interesting and challenging direction.
> We are afraid that this challenge may be out of the scope of this paper, because online settings are significantly different from what we consider in the paper.
> However, the first idea that comes to mind is to resort to dynamic clustering techniques for our landmark building.
> Due to the time constraint, we are unable to dive deep into the investigation.
> We thank the reviewer for bringing up this interesting direction, and we will explore it in the future.
>
> We hope that the reviewer will consider increasing the score.

---

> > ### Comment · Reviewer_WTuj · 2025-08-08
> > **Thanks for the rebuttal**
> >
> > The authors have addressed my concerns. I will raise my rating.

---

### Official Review · Reviewer_VE8S · 2025-07-03

**Clarity:** 2
**Significance:** 3
**Originality:** 3
**Rating:** 4
**Confidence:** 2

**Summary:**

The paper tackles the high cost of computing data Shapley values for KNN.
It introduces a “localized data Shapley” algorithm that limits valuation complexity to points within a small radius around each test example, found quickly via landmark-based clustering and a thresholded KNN search. The authors prove that, under mild perturbation-resilience assumptions, this strategy returns the exact Shapley values while cutting the runtime from quadratic to (nearly) linear in dataset size, and they show up to 25 times speed-ups on real datasets with up to a million points.

**Questions:**

1. Table 1 suggests mixed results on CIFAR-10/EMNIST (512-D). Could you add a controlled sweep over dimensionality (e.g., random projections) to isolate the “curse of dimensionality” effect?
2. Any practical heuristics for choosing τ in high dimensional spaces?
3. The paper focuses on thresholded KNN; can the landmark-based trick extend to weighted or learned-metric KNN (cv. Wang & Jia ’24)? A short discussion or a negative result would help.
4. How often does a real-world dataset satisfy the 3- (or 4-) perturbation-resilient k-center condition?
5. Could you include an empirical proxy (e.g., ratio of inter- to intra-cluster distances) or a stress-test that shows the algorithm’s degradation when the assumption is violated?

**Ethical Concerns:**

["NO or VERY MINOR ethics concerns only"]

**Final Justification:**

I reviewed the authors’ responses to my questions and comments, as well as to those from other reviewers, and have adjusted my rating accordingly.

**Limitations:**

Authors noted some technical limits, for example:
- Perturbation-resilience is hard to verify, but the authors justify it theoretically and show the method remains correct, falling back to baseline runtime if unmet.
- They note efficiency may taper in high dimensions => showed through results on CIFAR-10/EMNIST (512-D) datasets.
- They address τ and nL sensitivity with ablation plots and suggest using the “knee” point for tuning.

It would be nice to add a short section on how automated Shapley scores might exacerbate data-market inequities or enable privacy inference, and propose concrete safeguards (e.g., DP noise, audit trails).
Also discuss whether landmark clustering could leak sensitive structural info and how that risk can be mitigated in practice can show the depth of authors understanding of the potential impact of their works.

**Quality:**

3

**Strengths And Weaknesses:**

Strength
- Authors addressed a well-motivated and increasingly relevant problem: scalable data valuation.
- Introduced a novel algorithm combining threshold-based KNN with landmark-based clustering—both theoretically grounded and practically impactful.
- Offered a provable runtime guarantees under realistic structural assumptions (perturbation resilience).
- The empirical results showing up to 25× speedups over prior methods which was nice to have. However, I didn't have time to run their code, I might do that during the response period.

Weaknesses
- Some limitations in verifying perturbation resilience in practice are acknowledged, but not fully explored.
- Algorithm may degrade in high-dimensional settings due to curse of dimensionality—this isn’t deeply discussed.
- Slightly incremental in the sense that it builds heavily on prior KNN Shapley work, though the localization idea is novel.
– While the authors provide strong theoretical contributions, the experimental section could benefit from deeper analysis and broader exploration of dataset characteristics and more complex dataset to better demonstrate practical impact. Since the paper is submitted under general ML (not theory), a more robust experimental design is expected.

Not a weakness just a friendly suggestion: the citations currently start from [22] at the beginning of the paper, and it might be helpful to renumber them starting from [1]. This would make it easier to follow and track the references as we read through the paper.

---

> ### Author Rebuttal · Authors · 2025-07-28
>
> We thank the reviewer for reviewing our paper and many constructive comments.
>
>
> **Q1**:
> Table 1 suggests mixed results on CIFAR-10/EMNIST (512-D). Could you add a controlled sweep over dimensionality (e.g., random projections) to isolate the “curse of dimensionality”
> effect?
>
> **A1**:
> We perform projections by a Gaussian random matrix on the CIFAR-10 dataset to reduce its dimensionality to 10, and
> use the same ratio of $\tau/d$ as before.
> The baseline takes 116s (148s previously) and our method with $\tau/d=0.05$ takes 13s (75s previously).
> The running time of all methods decreases, due to the smaller overhead for distance computation, but the ratio of the running time of the baseline to ours increases.
> It seems effective to reduce the dimensionality by random projections.
> We thank the reviewer for the suggestion, and will add this discussion to the revised paper.
>
>
>
> **Q2**:
> Any practical heuristics for choosing τ in high dimensional spaces?
>
> **A2**:
> We are not aware of any good strategies for choosing $\tau$ in high-dimensional spaces.
> A general approach to choose $\tau$ is to tune it on a validation set for the task at hand (e.g., classification accuracy).
> We report simple statistics of the tuned ratio $\tau/d$ for datasets in our experiments for your reference.
> The unsurprising rule is that a larger $\tau$ usually leads to better accuracy.
> However, the marginal gain of increasing $\tau/d$ saturates at $0.05$ for 3/8 of the datasets (all large datasets) and at $0.2$ for all.
>
>
> **Q3**:
> The paper focuses on thresholded KNN; can the landmark-based trick extend to weighted or learned-metric KNN (cv. Wang & Jia ’24)? A short discussion or a negative result would
> help.
>
> **A3**:
> It is an interesting question about extending the technique to [JW24].
> As far as we know, they apply a very different dynamic programming (DP) technique to compute the kNN-Shapley values with respect to a "hard-label" utility function.
> Basically, they use DP to do counting of pivotal subsets for each data point.
> It seems possible to extend our technique to their setting by restricting the counting among data points whose distance to the test point is less than the radius $\tau$.
> We will explore this direction in the future, and thank you for your question.
>
>
> **Q4**:
> How often does a real-world dataset satisfy the 3- (or 4-) perturbation-resilient k-center condition?
>
> **A4**:
> Unfortunately, this condition rarely perfectly holds in real-world datasets.
> This is mostly due to the noise in the real-world data.
> For example, given two well-separated clusters with the majority of their data points satisfying the condition, there may be a small portion of outliers in each cluster that violate the condition.
> This is why we design the ball-expansion technique that searches for all data points within the radius $\tau$.
> It starts with the most likely data points and uses lightweight bounds to prune the search space, which allows it to adapt gracefully to the violations.
>
>
> **Q5**:
> Could you include an empirical proxy (e.g., ratio of inter- to intra-cluster distances) or a stress-test that shows the algorithm’s degradation when the assumption is violated?
>
> **A5**:
> We design a stress test with a random dataset.
> The dataset consists of two fixed centers surrounded by 5000 points each, which are created by sampling from a Gaussian distribution with the center being the mean.
> We gradually increase the variance to make the clusters less separated.
> Then, we run our method and monitor the change in its running time.
> It turns out that the running time of our method decreases.
> This is counter-intuitive to us at first sight.
> Later, we figure out that it can be beneficial to threshold-based KNN when the clusters become dispersed,
> as less points need to be considered during the landmark-based search.
> This is an interesting observation, and we thank the reviewer for the suggestion.

---

### Official Review · Reviewer_dXW1 · 2025-07-03

**Clarity:** 4
**Significance:** 2
**Originality:** 3
**Rating:** 5
**Confidence:** 3

**Summary:**

This paper considers the problem of efficiently computing Shapley values for a special class of functions. In general, Shapley values have $2^n$ terms, where $n$ is the number of Shapley values. However, for restricted classes of functions $v$, we can hope to compute the Shapley values more efficiently. Work by Jia et al. shows how to compute the Shapley value with respect to one test point for $k$ nearest neighbors in time $O(dn + n log n)$ where $d$ is the dimension of the points. However, scaling this calculation to $n$ test points would naively take $O(dnn + n^2 log n)$.

In this work, the authors build a "landmark" data structures that enables faster computation when the dataset is "stable" in a technical sense. They find experimentally that this method can perform better than prior works for computing the Shapley values.

**Questions:**

In table 1, each row corresponds to a different dataset. The baseline only has $\tau/d=.05$ but Algorithm 1 has $\tau/d =.05, .1, .2$. If you're going to vary $\tau/d$, it seems fair to also vary it for the baseline approach. Can you please run these experiments?

Regarding your discussion after Theorem 4, let's imagine $n_test \approx n$, and let's assume that the partitions are roughly equally sized so that $s \approx n / n_L$ (you could make this formal under the right distributional assumption with some log factors). Ignoring log factors, your bound is $O(n_L n d + n n/ n_L d)$. Optimizing over $n_L$, we would choose $n_L = \sqrt{n}$, then your bound is $O(n^{1.5 d})$. Would you present your result in this way? I think it makes the guarantee more digestible (with some mild assumptions), and makes it clear that you're getting a reasonable speed up from $n^2 d$ to $n^{1.5} d$.

Could you run experiments where you plot baseline vs algorithm 1 performance by size of dataset? It's hard to tell from the raw numbers how the *rate* of the difference is changing.

If a plot looks persuasive and you present your guarantee this way, I could change my opinion on the "marginal" impact of your work, and increase my score.

**Ethical Concerns:**

["NO or VERY MINOR ethics concerns only"]

**Final Justification:**

My concerns have been addressed about the speedup of their work. I'm not sure about the breadth of their impact, but the ideas are quite nice, and the algorithm works really well!

**Limitations:**

Yes.

**Quality:**

4

**Strengths And Weaknesses:**

Strengths:

* *Very* nice presentation of the method. The main ideas were all clearly and accessibly discussed.

* The main result that Shapley values can be more efficiently computed under a stability assumption is quite nice.

* The theoretical analysis and guarantees is a nice application of "perturbation resilience."

Weaknesses:

* The speed up seems somewhat marginal in practice? This could be an issue with the presentation, please see below in the question.

* To me, perhaps because I don't work in this exact space, the application of this work is somewhat limited.

---

> ### Author Rebuttal · Authors · 2025-07-28
>
> We thank the reviewer for the time and effort in reviewing our paper.
> We answer your questions below.
>
> **Q1**:
> In Table 1, each row corresponds to a different dataset. The baseline only has $\tau/d=0.05$ but Algorithm 1 has $\tau/d=0.05, 0.1, 0.2$. If you're going to vary $\tau/d$, it seems fair to also vary it for the baseline approach. Can you please run these experiments?
>
> **A1**:
> We didn't vary the parameter $\tau$ for the baseline because the running time of the baseline is less sensitive to $\tau/d$.
> More specifically, the baseline has to go through all the training points for each test point, regardless of the radius value $\tau$.
> Thus, its running time is at least $O(n_{train} \cdot n_{test})$, which is quadratic if the number of test points is large.
> Due to the space limit, we fix the radius $\tau$ to be the smallest value for the baseline in our experiments, which gives it an advantage over our method but not the other way around.
> We are happy to run these experiments with varying $\tau$ for the baseline, and include them in the appendix of the revised paper.
>
>
> **Q2**:
> Regarding your discussion after Theorem 4, let's imagine $n_{test} \approx n$, and let's assume that the partitions are roughly equally sized so that $s \approx n/n_L$ (you could make this formal under the right distributional assumption with some log factors). Ignoring log factors, your bound is $O(n_L nd + nn/n_L d)$. Optimizing over $n_L$, we would choose $n_L = \sqrt{n}$, then your bound is $O(n^{1.5} d)$.
> Would you present your result in this way? I think it makes the guarantee more digestible (with some mild assumptions), and makes it clear that you're getting a reasonable speedup $n^2$ to $n^{1.5}$.
>
> **A2**:
> Sure, it sounds more logical to present the result in this way.
> We are happy to revise the paper accordingly.
> Thanks for the suggestion!
>
>
>
> **Q3**:
> Could you run experiments where you plot baseline vs algorithm 1 performance by size of dataset? It's hard to tell from the raw numbers how the rate of the difference is changing.
> If a plot looks persuasive and you present your guarantee this way, I could change my opinion on the "marginal" impact of your work, and increase my score.
>
> **A3**:
> It is true, and we should have done this to help the readers digest the result.
> Unfortunately, we are not allowed to present images or external links during the rebuttal process.
> We have plotted the running time of the baseline and our method by the size of dataset ($n_{train} \cdot n_{test}$).
> Basically, the baseline is represented by a straight line that scales linearly with the size, while our method has ups and downs depending on the difficulty of the datasets.
> The line of our method is far below the baseline, and scales sublinearly with the size.

---

> > ### Comment · Reviewer_dXW1 · 2025-08-01
> >
> > Could you please present the new plot in a table or tables?
> >
> > Annoying about the ban on images.

---

> > > ### Author Response · Authors · 2025-08-01
> > >
> > > Sure, the table is given below.
> > >
> > > | size ($n_{train} \cdot n_{test}$) | runtime (baseline) | runtime (our-0.05) | runtime (our-0.1) | runtime (our-0.2) |
> > > |------|---------------------|----------------------|-------------------|-------------------|
> > > | 2.3e+06 | 1.2e+01 | 2.7e+00 | 3.2e+00 | 4.7e+00 |
> > > | 1.6e+07 | 8.6e+01 | 8.4e+00 | 1.2e+01 | 2.7e+01 |
> > > | 2.5e+07 | 1.5e+02 | 7.6e+01 | 8.7e+01 | 1.4e+02 |
> > > | 8.5e+07 | 4.5e+02 | 2.5e+02 | 2.7e+02 | 4.4e+02 |
> > > | 3.8e+08 | 2.0e+03 | 1.6e+02 | 2.5e+02 | 3.9e+02 |
> > > | 9.7e+08 | 6.1e+03 | 2.5e+03 | 3.6e+03 | 6.0e+03 |
> > > | 1.7e+09 | 1.0e+04 | 1.9e+03 | 4.1e+03 | 9.1e+03 |
> > > | 2.0e+09 | 1.2e+04 | 4.9e+02 | 9.8e+02 | 3.0e+03 |

---

### Official Review · Reviewer_x6A4 · 2025-07-03

**Clarity:** 3
**Significance:** 3
**Originality:** 3
**Rating:** 5
**Confidence:** 3

**Summary:**

The article addresses the problem of computing Shapley values for data valuation in K-nearest neighbor (KNN) models. By introducing a landmark-based near-neighbor search, the authors reduce the search space and significantly improve the computational efficiency of Data Shapley for threshold-based KNN. They analyze the theoretical complexity of their algorithm under a perturbation resilience assumption and support their findings with empirical validation on both synthetic and real-world datasets. The results demonstrate that the proposed methodology offers substantial improvements over classical KNN-based Shapley value computations.

**Questions:**

1. Are there known scenarios where perturbation resilience does not hold? If so, does your method still outperform the baseline in those cases?
2. You report improvements in execution time, but do not present any accuracy comparisons. How accurate is your method in approximating the Shapley values compared to the baseline?
3. How does your method perform against other state-of-the-art data valuation techniques—both in terms of computation time and accuracy?

**Ethical Concerns:**

["NO or VERY MINOR ethics concerns only"]

**Final Justification:**

The article is good and the authors replied to all my concerns during the rebuttals. I've dediced to keep my good score.

**Limitations:**

Yes

**Quality:**

3

**Strengths And Weaknesses:**

Strengths
- The article is clearly written and well-structured.
- The addressed problem—efficiently computing data Shapley values for KNN models—is relevant.
- The work is mathematically solid and theoretically grounded.
- The authors analyze the theoretical complexity of their method and provide sufficient conditions under which it improves upon existing time bounds in the literature.
- The proposed method shows strong empirical performance, outperforming the classical KNN Shapley approach.

Weaknesses
- The theoretical assumption on perturbation resilience is difficult to verify in practice.
- The empirical evaluation compares the proposed method only to the classical baseline, without including other competitive data valuation methods. Moreover, comparisons are done only with respect to execution time.

---

> ### Author Rebuttal · Authors · 2025-07-28
>
> We thank the reviewer for the time and effort in reviewing our paper.
> We answer your questions below.
>
> **Q1**:
> Are there known scenarios where perturbation resilience does not hold? If so, does your method still outperform the baseline in those cases?
>
> **A1**:
> Yes, perturbation resilience fails in scenarios such as (1) datasets with overlapping clusters where small perturbations can change cluster assignments, (2) high-dimensional data (i.e., curse of dimensionality) where distance-based clustering becomes ineffective, and (3) datasets with insufficient separation between clusters.
> However, even in these cases, our method still outperforms the baseline, though inevitably with a smaller speed-up.
> For example, see the results on high-dimensional datasets like CIFAR-10 and EMNIST in our experiments.
> Note that our worst-case complexity equals the baseline with negligible preprocessing overhead.
>
>
> **Q2**:
> You report improvements in execution time, but do not present any accuracy comparisons. How accurate is your method in approximating the Shapley values compared to the baseline?
>
> **A2**:
> We have added additional experiments about the accuracy of a popular downstream task.
> The short answer is that there is little difference between the accuracy of our method and that of the baseline kNN-Shapley.
> We refer the reviewer to our answer to reviewer WTuj for more details.
>
>
>
> **Q3**:
> How does your method perform against other state-of-the-art data valuation techniques—both in terms of computation time and accuracy?
>
> **A3**:
> The main focus of the paper is to accelerate the computation of kNN-Shapley, and to make the first attempt to address its scalability issue with respect to the number of test examples.
> We hope to stick to this goal and provide more focused comparisons for our purpose.
> The accuracy of our method is almost identical to the baseline kNN-Shapley, which should be unsurprising given their similar formulations.
> We refer readers to for example, the reference [19] for more comprehensive comparisons with other state-of-the-art data valuation techniques.

---

> > ### Comment · Reviewer_x6A4 · 2025-08-01
> >
> > I thank the authors for their answer. All my concerns have been clarified. I have no further questions.

---

### Decision · Program_Chairs · 2025-09-17

**Decision:**

Accept (poster)

**Comment:**

This paper looks at the fundamental question of data-valuation, using the Shapley value, which is not tractable in the general case. As a consequence, an important line of work consists in looking at subcases, with additional assumptions, and to derive algorithms that can perform it (or at least approximate the value with enough precision).

This paper does this, and all the reviewers, and myself, are rather positive about this paper, the writing and the contribution.

As a consequence, I am happy to recommend acceptance.